# miRNA Expression Analysis of the Hippocampus in a Vervet Monkey Model of Fetal Alcohol Spectrum Disorder Reveals a Potential Role in Global mRNA Downregulation

**DOI:** 10.3390/brainsci13060934

**Published:** 2023-06-09

**Authors:** Rob F. Gillis, Roberta M. Palmour

**Affiliations:** 1Department of Human Genetics, Faculty of Medicine and Health Sciences, McGill University, Montreal, QC H3A 0C7, Canada; robfgillis@gmail.com; 2Department of Psychiatry, Faculty of Medicine and Health Sciences, McGill University, Montreal, QC H3A 0G4, Canada; 3Behavioural Science Foundation, Mansion KN 0101, Saint Kitts and Nevis

**Keywords:** miRNA, prenatal alcohol, miRNA, hippocampus, non-human primate, miRNA–mRNA interaction

## Abstract

MicroRNAs (miRNAs) are short-length non-protein-coding RNA sequences that post-transcriptionally regulate gene expression in a broad range of cellular processes including neuro- development and have previously been implicated in fetal alcohol spectrum disorders (FASD). In this study, we use our vervet monkey model of FASD to follow up on a prior multivariate (developmental age × ethanol exposure) mRNA analysis (GSE173516) to explore the possibility that the global mRNA downregulation we observed in that study could be related to miRNA expression and function. We report here a predominance of upregulated and differentially expressed miRNAs. Further, the 24 most upregulated miRNAs were significantly correlated with their predicted targets (Target Scan 7.2). We then explored the relationship between these 24 miRNAs and the fold changes observed in their paired mRNA targets using two prediction platforms (Target Scan 7.2 and miRwalk 3.0). Compared to a list of non-differentially expressed miRNAs from our dataset, the 24 upregulated and differentially expressed miRNAs had a greater impact on the fold changes of their corresponding mRNA targets across both platforms. Taken together, this evidence raises the possibility that ethanol-induced upregulation of specific miRNAs might contribute functionally to the general downregulation of mRNAs observed by multiple investigators in response to prenatal alcohol exposure.

## 1. Introduction

Fetal alcohol spectrum disorder (FASD) is a conceptual term used to describe the entire range of adverse physiological and behavioural outcomes that stem from prenatal alcohol exposure (PAE) [1]. The modern clinical nosology of FASD began between 1968 and 1973 when two separate groups reported the effects of PAE in patient cohorts with CNS defects, growth retardation and a pattern of craniofacial dysmorphologies leading to the phenotypic characterization of fetal alcohol syndrome (FAS) [2,3]. It was gradually recognized that FAS did not capture the full range of phenotypic effects of PAE, leading to the blanket term FASD. In addition to the phenotypic presentations defined as FAS, individuals exposed to alcohol in utero may present with cardiac, skeletal, renal, ocular and auditory deficits [4,5,6,7,8,9] as well as long term social, emotional and behavioural impairments [10,11]. Risk factors for developing FASD involve the gestational timing of ethanol exposure [12], the extent (both quantity and duration) of ethanol exposure [13] and a genetic predisposition to the effects of PAE [14,15,16,17].

Establishing a prevalence for FASD is complicated by epidemiological and diagnostic uncertainties [18,19,20] such that estimates range from an average of 6.2/1000 in clinic-based studies to 38.2/1000 in active case ascertainment studies [21]. PAE is thus generally considered to be the leading cause of preventable developmental delay in the Western world. There have been many attempts to elucidate the origins of the altered developmental trajectory in FASD via gene expression studies [22,23,24,25,26,27,28,29,30,31,32] as well as higher-level regulation of gene expression, such as epigenetic modifications [33,34,35,36,37].

Given the role of miRNAs in the regulation of a diverse range of cellular processes such as development and apoptosis [38], neurogenesis and neurodegeneration [39,40,41], as well as their abundance in the CNS [42], they make interesting candidates in the exploration of developmental effects of PAE [43]. There have been multiple investigations of miRNAs in various PAE models, including neural stem cells [44], zebrafish [45] and mice [46,47]. There has also been consistent observation of miRNA upregulation as a consequence of ethanol exposure in both clinical [48,49] and preclinical [50,51] studies. Developing an understanding of how ethanol alters neurodevelopment will require amalgamation across multiple levels of regulation prior to developing effective intervention strategies.

The hippocampus is a particularly interesting structure to examine due to the well-documented effects that ethanol exerts in this vulnerable region. There have been many reports of PAE-influenced hippocampal cell deficits in rodents [52,53] and guinea pigs [54]. In vervet monkeys, the model for PAE employed in the current study, moderate ethanol exposure during the last half of pregnancy produced low cell count in CA regions of the hippocampus as well as a reduction in hippocampal volume [55]. Previous interrogation of hippocampal mRNA expression in this model revealed a global downregulation of gene expression when exploring alcohol as a main effect [56]. In the present study, we sought to determine if miRNA changes were also present and, if so, whether they might have been related to the mRNA fold changes in the prior study.

## 2. Materials and Methods

Animal care: Male vervet monkeys of two age groups [5-month-old infants (5.6 *±* 0.4 months); 2-year-old juveniles (26.4 ± 1.8 months)] as well as two phenotypes (PAE, control) were chosen for this study. Only male vervets (*Chlorocebus sabaeus*) were included in order to remove biological variance related to sex. The animals were housed in outdoor enclosures and kept within social groups under a captive breeding program managed by the Behavioural Science Foundation (Estridge Estate, Mansion, Federation of St Kitts and Nevis). This environment is designed to mimic natural living conditions and to provide regular foraging opportunities to the animals. Alcohol-preferring dams were used exclusively in this study to reduce the stress associated with alcohol administration in other studies and to avoid social interruption, they drank in individual compartments adjacent to their group cages. All animals were fed with High-protein Primate Chow (Harlan, KY, USA) as well as a mixture of local produce and fruits; access to clean drinking water was unlimited. All procedures in this study were reviewed and approved by the Animal Care Committees of McGill University (Montreal, QC, Canada, protocol #4627) and the Behavioural Science Foundation, both acting under the auspices of the Canadian Council on Animal Care.

Alcohol exposure: Each social group comprised several alcohol-preferring dams (defined as those that would voluntarily drink at least 2 g ethanol/kg body weight in a 4 h period) with a single alcohol-avoiding male. After group stability was achieved, the females were observed for evidence of menstrual cycling and reproductive behaviour; pregnancy was established via bi-weekly physical examination. Pregnant dams were shave-marked for rapid identification at approximately the midpoint of the modal 165-day gestation period. Thereafter, they drank 4 days per week (M, Tu, Th, F) for a period of 4 h/day beginning at about 8:30 a.m. As mentioned above, each animal drank in an individual compartment adjacent to her social group. During the alcohol administration period, PAE dams were given the option to drink 8% beverage ethanol in tap water, while control dams were offered an equal volume of isocaloric sucrose with no alcohol. Water was available ad libitum throughout the drinking period. Volumes of the ethanol solution were individually varied by the weight of each animal to enable each PAE mother to ingest up to 3.5 g ethanol/kg body weight in a single session. After the 4 h exposure period, all animals were returned to their social groups. To ensure that the alcohol was physically ingested, a small amount of blood was collected from unanesthetized dams bi-weekly for measurement of blood alcohol concentration. These measurements are available in the Appendix A.

Tissue Collection: As the sacrifice data approached, age-matched cases and controls were moved from social groups to individual cages about 4 days in advance. This step was intended to reduce the non-biological variance that group dynamics could provide. Sacrifices were performed between 10 a.m. and noon to reduce variance related to circadian rhythm. Animals were assigned a random sacrifice order alternating between cases and controls and were euthanized according to an ACUC-approved standard operating procedure. Briefly, animals were sedated with 10 mg/kg ketamine/xylazine (9:1 *v*/*v*) then terminated with a 3 mL intravenous injection of pentobarbital solution (20% Somnotol *w*/*v*). Lungs were deflated once respiration and cardiac contractions ceased [57]. The rapidly extracted brains were perfused with ice-cold RNase-free phosphate-buffered saline (PBS) then dissected with sterile instruments rinsed in RNase-free PBS. A sagittal cut was performed to separate the left and right hemispheres. The left hippocampus was then removed and transferred to a sterile vial containing 1 mL guanidine isothiocyanate/phenol solution per 100 mg tissue, according to the manufacturer’s protocol (Quizol, Qiagen, Germany). The hippocampus was immediately homogenized using a portable homogenizer (Tissue Master 125; Omni, Rochester, NY, USA). Aliquots of 1 mL were transferred to sterile cryovials, frozen on dry ice and maintained at −40 °C prior to dry-ice shipping to Montreal for processing and analysis.

RNA Preparation: The Qiagen miRNeasy kit (Qiagen, Germany) was used to extract total RNA under RNase-free conditions following the manufacturer’s protocol. Quality analysis and microarray hybridization of these samples were performed at the McGill and Genome Quebec Innovation Centre (Montréal, QC, Canada). The concentration of RNA (mRNA and miRNA) was measured using the NanoDrop ND-1000 spectrophotometer (Thermo Scientific, Waltham, MA, USA), and the quality of the RNA was analyzed using the Bioanalyzer 2100 (Agilent, Santa Clara, CA, USA). Furthermore, cRNA quality was also analyzed with the Agilent Bioanalyzer, and all cRNA samples used in the final study passed this stage of quality control. In total, 32 male vervet RNA samples were screened with 24 selected for final array hybridization based on preferred sample quality to create four groups with 6 replicates per group (5-month control, 5-month PAE, 2 yr control, 2 yr PAE).

Array Hybridization, Quality Control, Analysis: All samples were hybridized to Affymetrix GeneChip miRNA 3.0 arrays on the same day using the same technician. They were assigned a random loading order unrelated to biological grouping. The raw data were normalized using the robust multi-array average (RMA) normalization method [58]. These data were also explored further with the FlexArray software package [59]. All arrays passed quality control; however, as evidenced by the boxplots for the 24 arrays (Appendix A) one 2 yr old control (CONT2_4) had a lower mean expression level, skewing the entire group downward, and was thus eliminated from further analysis. Principal components analysis (Appendix A) showed that none of the remaining arrays were outliers. The final samples thus had an unbalanced 2 × 2 design with 23 arrays.

Probe Filtering: The GeneChip miRNA 3.0 comprises 25,016 miRNAs across several species. Although not designed specifically for the vervet monkey, miRNAs across the primate lineage are highly conserved [60]. We began filtering by using MAS 5.0 Absolute Detection function in the Affymetrix power tools software suite to select miRNAs. Only probe sets that showed expression in at least 6 arrays (*p* value < 0.05) were retained for further analysis. This step reduced the miRNAs available for analysis to 6288. Then, we exclusively selected miRNAs derived from humans, apes and Old World monkeys, reducing the number of expressed miRNAs further to 1756. This list included multiple probe IDs across several species that corresponded to the same molecular sequence. We therefore pared this list further to produce a unique miRNA list corresponding to the human miRbank ID, leaving a total of 613 expressed and unique miRNAs for interrogation. We then downloaded the updated miRNA names and corresponding sequence from miRbase [61] and matched them to the corresponding names and sequences from the miRNA 3.0 gene chip and subsequently used the most recent human miRNA nomenclature. The data, nomenclature and sequences used to annotate our miRNAs can be found within the Appendix A section.

Statistical Analysis: Consistent with the research design, the retained datasets were analyzed using a two-factor analysis of variance (ANOVA) with two main factors (age, alcohol), each with two levels (5 months/2 years; PAE/Control). A total of 24 animals were selected for final array hybridization with 6 replicates per group (5-month control, 5-month PAE, 2 yr control, 2 yr PAE). After elimination of one 2 yr control due to poor-quality array data, the 2 yr control group was reduced to 5 replicates. The *p*-values were subsequently converted to *q*-values to control for multiple testing [62].

Target Correlation Analysis: After identification of a list of 27 differentially expressed miRNAs (DEmiRNAs), an analysis of their mRNA targets was performed to interrogate the relationship between these DEmiRNAs and the pattern of generalized downregulation observed in our transcriptomic (mRNA) dataset. At this stage, miRNA target predictions were obtained using the TargetScan 7.2 algorithm [63]. The targeted mRNAs were then ranked according to the number of times they were independently targeted by the 24 upregulated miRNAs; this was further explored via three principal avenues. First, we performed correlation tests between the DEmiRNAs and their predicted targets using the cor.test function in R 4.02, a language and environment for statistical programming (www.R-project.org, accessed on 22 November 2021) [64]. Second, we measured the impact that combinatorial regulation of these predicted targets of our DEmiRNA list had on the fold change of the miRNA targets in our corresponding mRNA dataset. We performed this by tallying the number of independent times these mRNAs were predicted to be targeted by our 24 DEmiRNAs. Independent target count was then used as a classifier between 0 and 24 (times targeted by the DEmiRNAs), and the mean fold change in the mRNA targets was calculated for each group. Last, we generated a corresponding sample of 24 non-differentially expressed miRNAs with the highest *p*-values (i.e., approaching 1) in the analysis of alcohol exposure as a main effect, identified these as non-DEmiRNAs and measured the fold changes they induced in their corresponding mRNA targets. We analyzed the fold change difference between each classification group (DEmiRNA and non-DEmiRNA) in order to further narrow the role that DEmiRNAs could have played in relationship to the decreased fold changes in our corresponding mRNA dataset. Note that, in compiling this list of non-DEmiRNAs, we excluded any miRNA that showed a strong interaction effect.

## 3. Results

We provide here a tree diagram (Figure 1) to facilitate the progression of the reader through our results section.

### 3.1. ANOVA

#### 3.1.1. Main Effect: Alcohol

When interrogating alcohol exposure as a main effect in our 2 × 2 ANOVA design, we observed a departure from the null hypothesis with 118 miRNAs displaying a *p*-value less than 0.05, as opposed to 30 miRNAs that would have been expected if this independent variable had no impact on miRNA expression. The histogram of *p*-values in Appendix A demonstrates a clean ascension toward the significance threshold. We also observed a roughly equal proportion of total miRNAs that were upregulated (298) vs. those that were downregulated (315). However, there was an overabundance of upregulated miRNAs with *p* < 0.05 and log2 fold changes greater than 1 as can be observed in the volcano plot of Figure 2.

A total of 27 miRNAs emerge from this analysis that each have a *p*-value < 0.05 (*q*-value < 0.113) and a log2 fold change >1 or <−1; 24 of these were upregulated, while 3 were downregulated. The full list of differentially expressed miRNAs and their *q*-values is displayed in Appendix A.

#### 3.1.2. Main Effect: Age 

The 2 × 2 experimental design also allowed us to examine age as a main effect. This analysis also produced an overabundance of significant *p*-values with 97 miRNAs as opposed to the 30 that would be expected under the null hypothesis. Using age as a main effect, the histogram of *p*-values (Appendix A) also showed a smooth ascension toward the significance threshold. The proportion of upregulated vs. downregulated genes (339 up, 274 down) between age 5 months and age 2 years (using 5 months as the point of comparison) was relatively balanced. Figure 3 displays the relationship between fold change and *p*-value using age as an independent variable, and Appendix A lists miRNAs differentially regulated with respect to age.

#### 3.1.3. Interaction of Alcohol × Age

We also found an interaction effect with 250 miRNAs displaying *p*-values < 0.05 as opposed to the 30 that would have been expected under the null hypothesis. Further examination of these differentially expressed miRNAs with their interaction plots (Appendix A) reveals that the miRNAs with significant interaction effects do not display a single generalizable interpretation. Instead, multiple types of interaction plots are possible, and the interpretation is dependent upon the miRNA in question. An expanded list of the miRNAs with the lowest *p*-values for interaction are available in Appendix A.

### 3.2. Integrated Analysis miRNA–mRNA

#### 3.2.1. miRNA–mRNA Target Correlations

The predicted mRNA targets for a selected group of 24 upregulated miRNAs when exploring alcohol as a main effect were used to determine if the miRNAs could have played a direct role in the global downregulation of mRNA expression levels that we observed in a previous transcriptomic study of the same samples (GeneChip Rhesus Macaque Genome Array, [56]). The miRNAs defined as differentially expressed for this stage of the analysis had a *p*-value < 0.05 and a log2 fold change greater than 1. The first step in the integrated analysis was to download the target predictions for these 24 differentially expressed miRNAs (DEmiRNAs) using TargetScan 7.2 [63]. This collectively resulted in 8212 target mRNAs for these 24 upregulated miRNAs. The expression levels of these miRNAs were then correlated to their specific mRNA targets using the cor.test function in R v4.05, which can test for correlations between paired samples (http://www.R-project.org, accessed on 22 November 2021) [64]. An individual correlation profile was generated between each miRNA and its targets which included their distribution and *p*-values (available upon request). The summary of the *p*-values between all 24 miRNAs and their specific targets using TargetScan 7.2 are presented below in Figure 4.

In Figure 4 we see a clear excess of significant positive and negative correlations between these DEmiRNAs and their predicted mRNA targets. On average, there appears to be a slightly greater number of positive vs. negative correlations. However, if we closely examine the more significant fraction of miRNA–mRNA correlations with *p*-values < 0.0025 and divide this cross section of *p*-values further into twenty equal-sized bins, we see that the trend of a greater number of highly significant *p*-values remains true (upward trend toward the right), but we now observe a greater number of negative correlations than positive correlations (Figure 5).

At this stage in the analysis, we can conclude that we have an excess of upregulated miRNAs, an excess of downregulated mRNAs and now an excess of significant correlations between the paired samples with a trend toward a greater number of inverse correlations as the significance level of these correlations increases. An excess of both positive and negative correlations between miRNAs and their target mRNAs is not unexpected and is not proof of the direct involvement of miRNAs in the downregulation of mRNAs; however, this does display the existence of transcriptional networks and a functional link between these two paired datasets. In fact, without an excess of significant correlations here, it would be difficult to justify the pursuit of a functional link between these two datasets.

#### 3.2.2. Differentially Expressed miRNAs and mRNA Fold Changes

We next sought to further the evidence for a significant link between the global downregulation observed in our previous mRNA dataset [56] with the predicted mRNA targets (Target Scan 7.2) of the upregulated DEmiRNA gene list. Given that miRNAs can display combinatorial regulation [65] as well as the prediction that as any given mRNA is likely to be targeted by multiple miRNAs, some of these may be true functional relationships. We therefore explored the relationship between mRNA fold change and the number of times these mRNAs were individually targeted by our DEmiRNA list (Figure 6).

When analyzing the fold changes in this manner, we observed an inverse correlation between the fold changes of our mRNAs and the number of individual target predictions from our DEmiRNA list. The *p*-value of this correlation coefficient between the fold change in the mRNA and the number of times an mRNA was targeted is 1.79 × 10^−44^. It should be clarified here that we are not referring to total predicted sites; we are referring to the number of times each was targeted by one of the miRNAs in our DEmiRNA list. As we can observe in Figure 6, as the number of times an mRNA is targeted by an miRNA from the DEmiRNA list increases, the mean fold change of that group trends downward.

At this stage in the analysis, we can reasonably hypothesize that the increased number of significant correlations observed in Figure 4 and Figure 5 may actually translate into the downregulation of mRNA expression observed in our earlier study of the effects of PAE on hippocampal transcription. However, this functional link cannot solely be attributed to the DEmiRNA list from this study given that there is a strong overlap between mRNA predicted targets in all miRNAs. As a further test of the specificity of this relationship, we then downloaded the mRNA target predictions for the list of 24 non-DEmiRNAs chosen on the basis of *p*-values approaching 1 and described in further detail in the methods section (target correlation analysis). There was a strong degree of overlap between the non-differentially expressed and the differentially expressed miRNA–mRNA targets using TargetScan 7.2. The Pearson correlation coefficient between the number of times an mRNA was targeted by our DEmiRNA list and our corresponding non-DEmiRNA list was 0.72. This can be interpreted to mean that if a gene was frequently targeted by our DEmiRNA list, it was also a frequent target of a corresponding list of non-DEmiRNAs and might lead to the supposition that miRNAs in general may have a functional role in the downregulation of probes in our mRNA dataset without attributing a specific functional link to the DEmiRNA list. If however, the DEmiRNA list contributed directly to the downregulation observed in the parallel PAE mRNA dataset, one would predict that the mean residual mRNA fold change for each category (# times probe targeted by miRNAs in the DEmiRNA list vs. the non-DEmiRNA list) would be negative; i.e., there should be a greater number of functional miRNA–mRNA targets in our differentially expressed vs. non-differentially expressed miRNA targets despite the strong overlap in target prediction between the two lists. The result of this analysis is visible below in Figure 7.

In Figure 7, we see that the residuals (DEmiRNA targets and non-DEmiRNA targets) are largely negative for the majority of categories (# of times targeted), and indeed the mean fold change using alcohol as a main effect is lower for the list of DEmiRNA targets with four or more hits vs. the corresponding list of non DEmiRNA with four or more hits (−0.034 vs. −0.024, two tailed *t*-test, *p* = 4.16 × 10^−6^).

There is a considerable level of variation in the results produced by various miRNA target prediction algorithms, and overall agreement from one platform to the next can often be quite low [66]. Given this degree of variation between algorithms regarding target prediction, it could be argued that this effect was only observed due to an insidious artifact of the TargetScan 7.2 algorithm and the manner in which this software generates miRNA-mRNA target predictions. Therefore, this process was replicated using a different tool; miRWalk 3.0 [67]. miRWalk 3.0 is a user-friendly platform that scans the entire gene from the 5′ leader sequence through the coding regions as well as the 3′ untranslated region to identify miRNA target sequences throughout the length of the gene [66]. Figure 8 illustrates the replication of the trend observed with TargetScan 7.2 using miRWalk 3.0.

We also observed an inverse correlation between fold change in the downregulated PAE-related mRNAs and the number of times it was targeted by an individual miRNA (*p* = 6 × 10^−14^) using miRWalk 3.0 [67]. Interestingly, the mRNAs that were predicted not to have been targeted at all (0 times targeted) using miRWalk 3.0 had lower fold changes than the mRNAs that were predicted to be targeted up to four times. It should be noted however that we also performed a similar analysis using the miRanda algorithm [68], and the mean fold change for genes that were not predicted to be targeted by any of the DEmiRNAs was positive (Appendix A), indicating that the choice of miRNA prediction software does influence the outcome. Nonetheless, the trend of more targets equals greater fold change remains true across all algorithms tested. We were also able to replicate the negative residuals obtained by subtracting the fold changes of DEmiRNA targets from the fold changes of non-DEmiRNA targets obtained with miRWalk 3.0 (Figure 9).

We then generated a correlation plot while ordering the 24 miRNAs according to our first principal component to identify if there were any striking expression patterns amongst our DEmiRNA list. The results are shown in Figure 10.

As can be observed in the correlation matrix shown in Figure 10, the DEmiRNAs all have a strong positive correlation in their expression profiles ranging from 0.5 to 0.99. Aligning the probe sequences for the DEmiRNAs with the vervet genome identified a cluster of miRNAs (miR-539-5p, miR-376-3p, miR-495-3p, miR-543 and miR-329-3p) that form an expression cluster on vervet chromosome 24. These variants have a mean correlation coefficient of 0.97 as a sub-group, indicating the potential for combinatorial regulation of mRNA expression levels.

## 4. Discussion

This study explored the role that miRNAs might play in the development of FASD using a non-human primate (*Chlorocebus sabaeus*) model of prenatal alcohol exposure. Three focused identifications resulted: (1) we identified a list of 27 differentially expressed miRNAs (24 upregulated, 3 downregulated); (2) we confirmed the upregulation and differential expression of miR-9 which has frequently been implicated in other models of fetal ethanol exposure; and (3) we documented a plausible functional connection between the 24 DEmiRNAs identified in this study and the downregulated mRNAs reported in a previous transcriptomic study of these paired samples, collected concurrently. Each of these results is discussed below.

When exploring this dataset using alcohol as a main effect, we identified 24 upregulated miRNAs and 3 downregulated miRNAs. The excess of upregulated miRNAs in response to ethanol exposure is not an isolated finding [46,47,48,49,50], which makes our results consistent with prior studies. Given that gene expression studies are limited to specific temporal comparisons with age-matched controls, it could be theorized that this frequently observed upregulation of differentially expressed miRNAs simply stems from a general developmental delay induced by the stress of prenatal ethanol exposure. It is possible that the expression level of these miRNAs would decrease naturally in age-matched controls to elicit or induce a developmental transition yet remain elevated for longer periods in monkeys exposed to alcohol in utero, the effect of which would disappear using different temporal comparisons. This could explain why this effect is not universal across all miRNA studies to date [46,69]. Long-term measurements to determine if mRNA levels eventually regress to the mean are not yet available, but contrasting longitudinal studies of humans with alcohol dependence do show upregulation of brain miRNAs [49], while rats with alcohol dependence display a relative balance of upregulated and downregulated miRNAs [70]. It is thus an open question as to whether the observed upregulation of miRNA represents a byproduct of neurodevelopmental delay or a direct functional relationship between ethanol and miRNA expression.

Perhaps the miRNA of greatest interest in the development of FASD has been miR-9-5p (formerly known as miR-9). This miRNA is one of the 24 upregulated DEmiRNAs reported here; it has also been shown to be differentially expressed in at least eight different studies regarding ethanol exposure and miRNA expression (reviewed in [71]). This candidate miRNA has multiple roles in neurodevelopment including stem cell proliferation and maturation, neurogenesis and boundary formation [72,73,74]. Expression levels of miR-9-5p are consistently downregulated after neural progenitor cell exposure to ethanol but are upregulated in studies interrogating later stages of brain maturity and development [71].

A recent meta-analysis of gene expression in animal models of FASD identified 104 mRNAs that showed strong evidence of differential expression [75]. Remarkably, all 104 signature genes were downregulated. This general downregulation was also observed in our paired mRNA dataset [56], with 618 of 938 genes with a *p*-value < 0.05 being downregulated, while 320 were upregulated. As described in this communication, concurrently collected samples showed upregulation of miRNAs differentially expressed in PAE vs. control subjects. Although the data were obtained on different platforms (GeneChip miRNA 3.0, Rhesus Macaque Genome Array for mRNA), the potential functional relationship between the datasets was intriguing and led us to further explore this avenue.

Studies that have thus far been able to identify a direct relationship between upregulation of a specific miRNAs and downregulation of specific mRNA are sparse, and most rely on explanted tissue or ex vivo cell preparations (reviewed in [76]). Identifying a specific causal effect of this relationship will likely require gene manipulation, a functional assay and extensive molecular biology. However, with a paired miRNA–mRNA dataset, multiple lines of inquiry can be combined to test the evidence of a functional relationship between DEmiRNAs and their predicted targets. We began by correlating the expression of differentially expressed miRNA profiles with their predicted mRNA targets using TargetScan 7.2 and discovered an excess of significant correlations. Interestingly, in total, there are a greater number of positive correlations, but this pattern reversed to a predominance of negative correlations as the *p*-value of the correlation decreases. The observed positive correlations between miRNAs and their predicted targets are common and can indicate the presence of positively regulated functional miRNA–mRNA regulatory relationships. The reversal of this characteristic in the data at the most significant *p*-values indicates, however, the potential that these DEmiRNAs are influencing the expression levels of their true target pairs.

We then explored the impact that combinatorial regulation from our DEmiRNA list might have on the fold changes of their corresponding mRNA targets and were able to show a significant impact on these fold changes that increased as these mRNAs were targeted in a combinatorial manner. We showed a significant difference in the residual fold change after subtracting fold changes induced by a non-differentially expressed gene list from those hypothetically induced by a differentially expressed gene list then replicated this finding using a different target prediction algorithm.

The global prevalence of FASDs is estimated to be at least 0.8% [77], and some authorities report that this represents less than half of all true cases [78]. There is no specific medical treatment for FASD, but medications commonly used for the treatment of childhood hyperactivity, impulsivity and anxiety (all of which are exhibited by some children with FASD, and all of which are related to hippocampal function) may be useful [79]. Early diagnosis and supportive cognitive and behavioural training are clearly indicated [77]. With respect to the subject of this communication, a recent clinical trial identifying miRNA signatures in the blood of affected individuals may be particularly important [48]. This tool could potentially be used not only to improve early diagnosis of FASD but also as a probe for prenatal identification and treatment.

## 5. Conclusions

To our knowledge, this is the first study to observe the simultaneous upregulation of miRNA with correlated downregulation of mRNA in response to prenatal ethanol exposure using paired, concurrently sampled datasets. This is also the first example in which a hypothetical functional impact of global miRNA changes on global old changes in mRNA has been robustly investigated.

## Figures and Tables

**Figure 1 brainsci-13-00934-f001:**
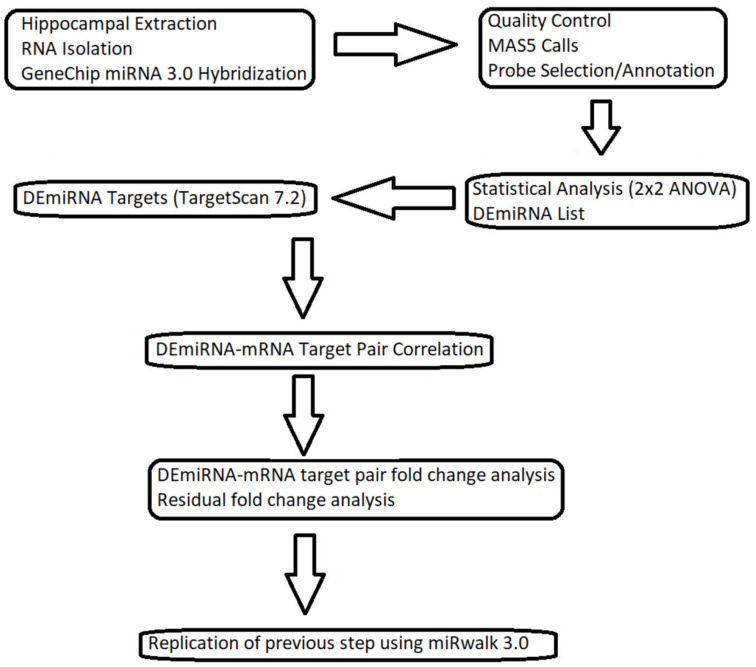
Flow chart describing the methodology to guide the reader through our results section.

**Figure 2 brainsci-13-00934-f002:**
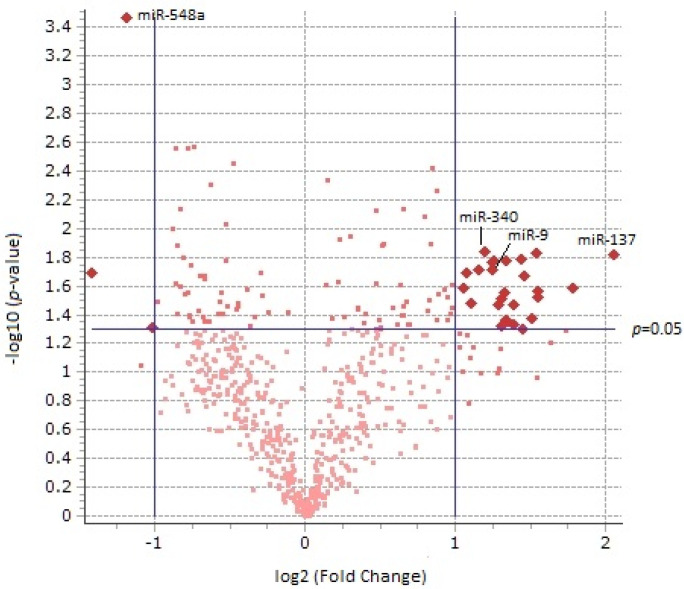
Volcano plot displaying the relationship between fold change (log_2_) and *p*-value (−log_10_) using alcohol as a main effect between pregnant vervet dams that voluntarily consumed alcohol between e100-165 and their sucrose-matched controls. There was an overabundance of differentially expressed up-regulated miRNAs vs. downregulated miRNAs exploring alcohol as a main effect. Twenty-seven miRNAs revealed both a *p*-value < 0.05 and a log2 fold change >1.

**Figure 3 brainsci-13-00934-f003:**
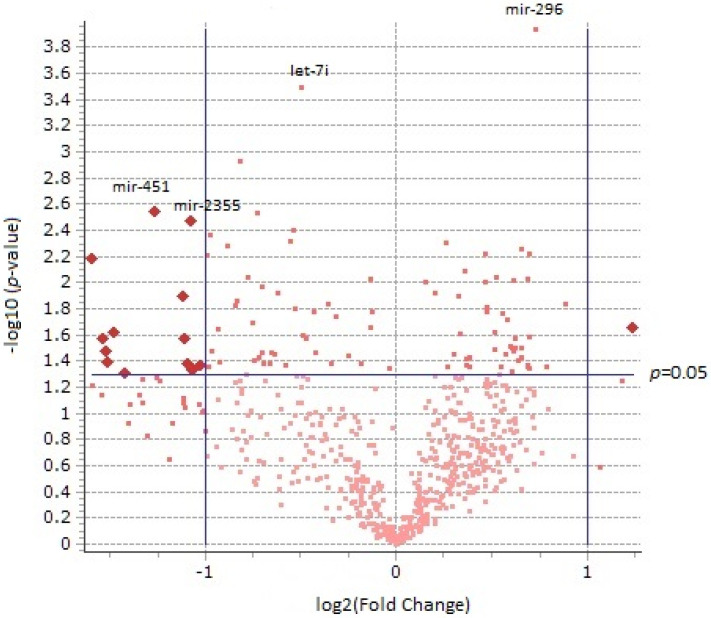
Volcano plot displaying relationship of *p*-values with fold changes using age as a main effect in 619 miRNAs. Each dot represents a miRNA which has been plotted against the −log_10_ of its *p*-value (vertical axis) and log_2_ fold change (horizontal axis) contrasting its expression value at 5 months and 2 years.

**Figure 4 brainsci-13-00934-f004:**
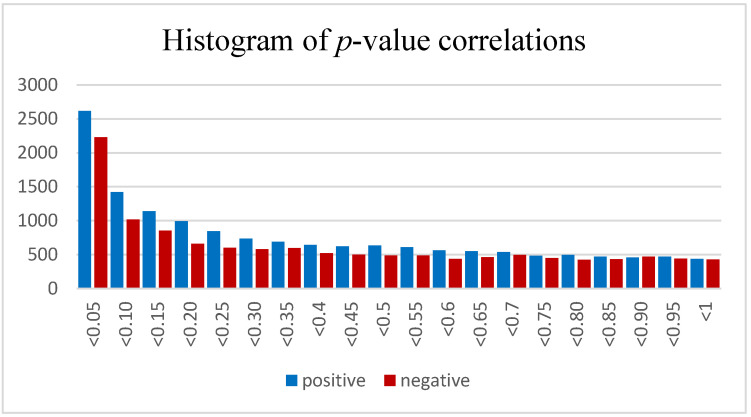
Histogram of *p*-values for correlations between 24 upregulated and differentially expressed miRNAs identified using alcohol as a main effect and their individual mRNA targets that were predicted using TargetScan 7.2. Each bin within the histogram represents the sum of correlation pairs corresponding to a range within 0.05 with both positive and negative correlations, coloured in blue and red, respectively.

**Figure 5 brainsci-13-00934-f005:**
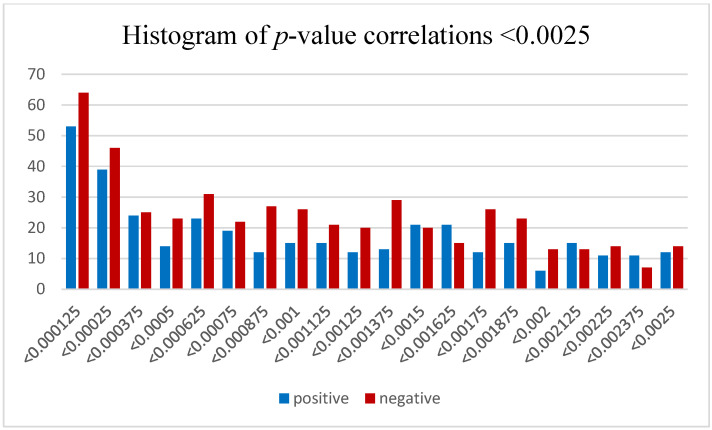
Histogram of *p*-values <0.0025 for correlations between 24 upregulated and differentially expressed miRNAs identified using alcohol as a main effect and their individual mRNA targets that were predicted using TargetScan 7.2. Each bin represents the sum of correlation pairs with both positive and negative *p*-values with a linear decrease of 0.0025/20 to create twenty equally sized bins between 0 and 0.0025. These have been separated to display positive and negative correlations in blue and red, respectively.

**Figure 6 brainsci-13-00934-f006:**
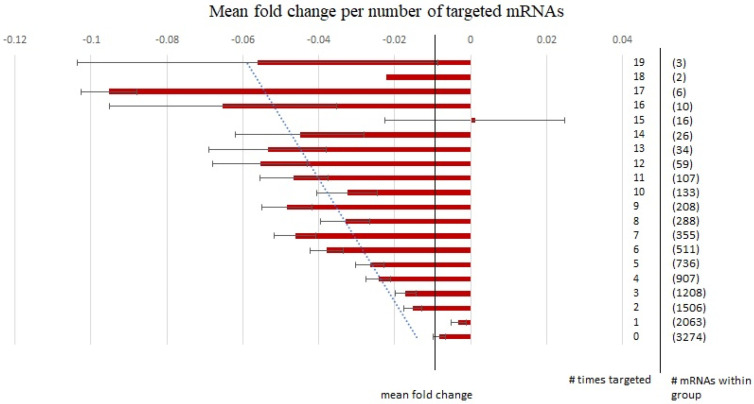
The mean fold change of each mRNA corresponding to the number of times it was targeted by one of the 24 upregulated DEmiRNAs identified using alcohol as a main effect as predicted by TargetScan 7.2. The vertical line labelled “mean fold change” represents the average fold change of all mRNAs using alcohol as a main effect, # times targeted ranges from 0 to 19 and represents the number of times a given mRNA was targeted by our DEmiRNA list and # mRNAs within group represents the number of mRNAs within each target group for which the mean fold change was calculated. Error bars were calculated using the standard deviation of the mean.

**Figure 7 brainsci-13-00934-f007:**
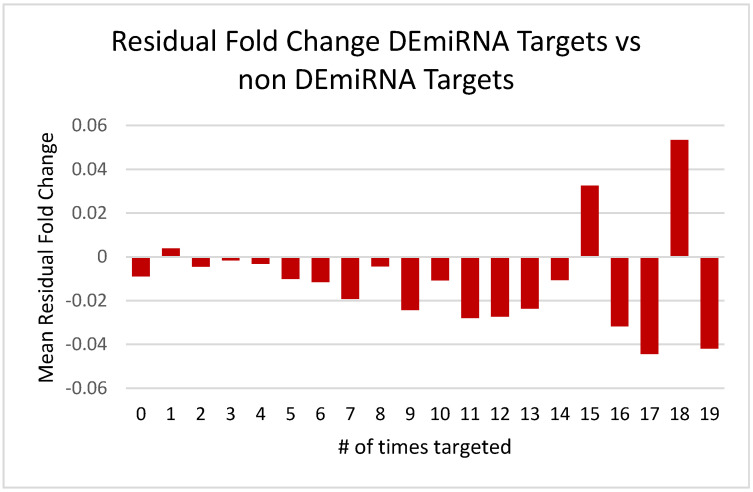
Residual fold changes (DEmiRNA and non-DEmiRNA) for each grouping of mRNAs based on the number of times they were targeted by miRNAs from each category using alcohol as a main effect. The miRNA–mRNA target pairs were predicted using TargetScan 7.2 for Figure 7.

**Figure 8 brainsci-13-00934-f008:**
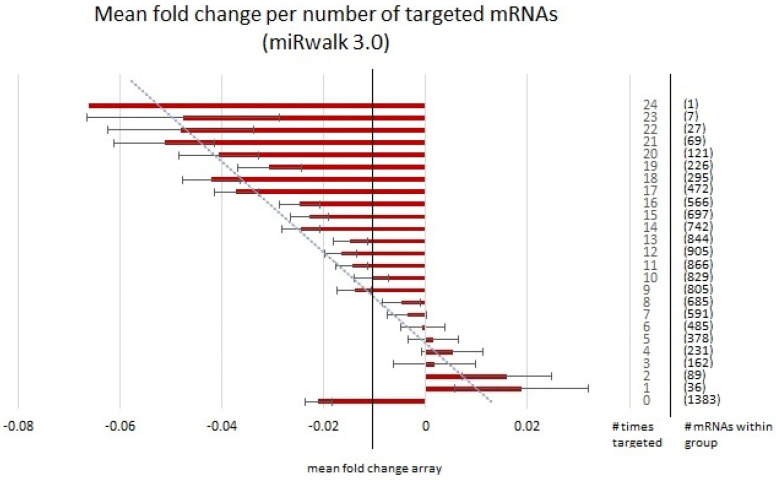
Graph representing the mean fold change of each mRNA corresponding to the number of times it was targeted by our list of 24 upregulated DEmiRNAs using alcohol as a main effect as predicted by miRwalk 3.0 as a replication algorithm. The vertical line labelled “mean fold change” represents the average fold change of all mRNAs using alcohol as a main effect, # times targeted ranges from 0 to 24 and represents the number of times a given mRNA was targeted by our DEmiRNA list and # mRNAs within group represents the number of mRNAs within each target group for which the mean fold change was calculated. Error bars were calculated using the standard deviation of the mean.

**Figure 9 brainsci-13-00934-f009:**
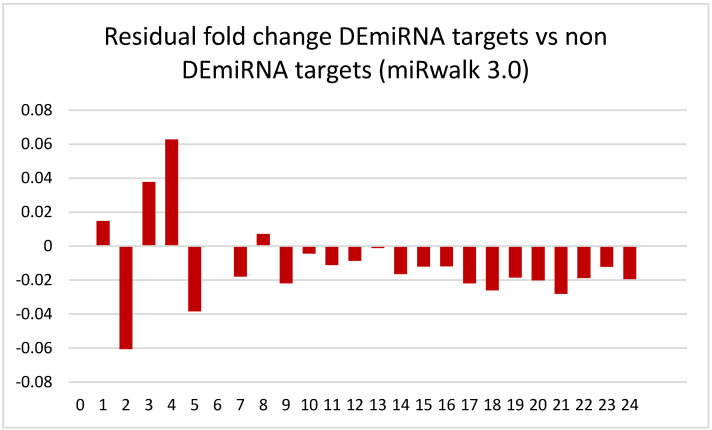
Residual fold changes (DEmiRNA and non-DEmiRNA) for each grouping of mRNAs based on the number of times they were targeted by miRNAs from each category using alcohol as a main effect with miRNA–mRNA target pairs predicted in both miRNA gene lists using miRwalk 3.0.

**Figure 10 brainsci-13-00934-f010:**
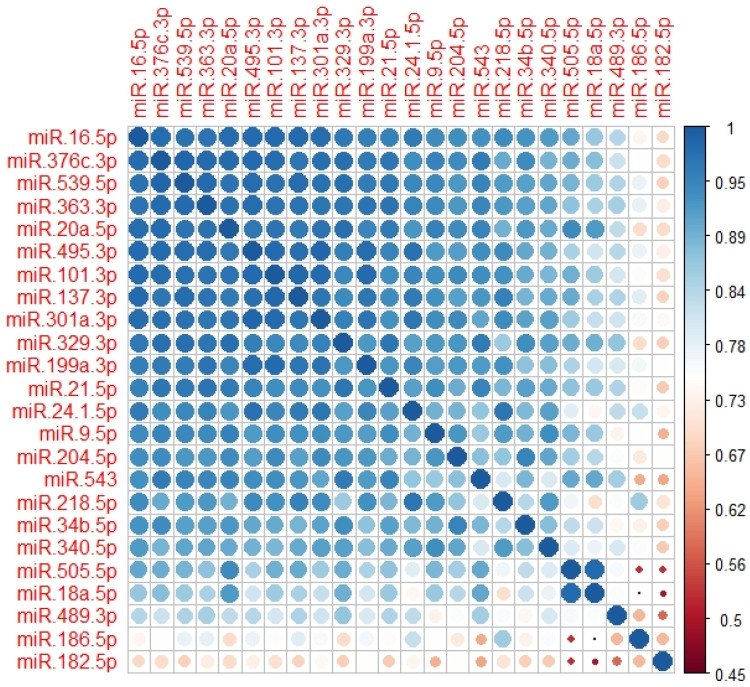
Plot of a correlation matrix for our list of 24 upregulated DEmiRNAs obtained in a 2 × 2 ANOVA using alcohol as a main effect. DEmiRNAs are ordered by first principal component. The size of the circle and intensity of the colour represent the strength of correlation. The legend has been re-scaled from 0.45 to 1 for greater sensitivity.

## Data Availability

All data that were used to generate these results have been deposited with the gene expression omnibus at the National Institute of Health under the accession number GSE181456. Data related to the parallel transcriptome study referred to at length in this manuscript are also stored at NIH under the accession number GSE173516. Additional information regarding the samples can also be accessed through the datadryad organization using the following url: https://datadryad.org/stash/share/ej24eEr73GZTbOuWrEcArSvQdnGi48Q2WNym4V6ea7Y, accessed on 3 March 2022.

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
