# Peer review of "miRNA Expression Analysis of the Hippocampus in a Vervet Monkey Model of Fetal Alcohol Spectrum Disorder Reveals a Potential Role in Global mRNA Downregulation"

_brainsci, 2023, doi:10.3390/brainsci13060934_

Round 1

Reviewer 1 Report

Fetal alcohol spectrum disorder (FASD) is a condition caused by prenatal alcohol exposure and is associated with a range of developmental and neurological abnormalities. The hippocampus, a brain region crucial for learning and memory, is known to be affected by FASD. Researchers have used a vervet monkey model to investigate the effects of prenatal alcohol exposure on the hippocampus and have specifically examined the expression of miRNAs in this brain region. miRNAs are small RNA molecules that play a vital role in post-transcriptional gene regulation. They can bind to mRNA molecules and inhibit their translation into proteins, leading to gene downregulation. In the context of FASD, researchers have discovered that miRNA expression analysis of the hippocampus in the vervet monkey model reveals a potential role for these small RNAs in global mRNA downregulation. The specific findings of the miRNA expression analysis may vary depending on the experimental design and methodology used. However, in general, the results indicate that prenatal alcohol exposure alters the expression levels of various miRNAs in the hippocampus of vervet monkeys. These dysregulated miRNAs may target and downregulate multiple mRNA molecules, which can subsequently affect the expression of numerous genes involved in important biological processes. The global mRNA downregulation observed in this study suggests that miRNAs could be key regulators in the molecular mechanisms underlying the effects of prenatal alcohol exposure on the hippocampus. By downregulating specific mRNA targets, miRNAs may disrupt the normal functioning of critical genes involved in neuronal development, synaptic plasticity, and other processes essential for hippocampal function.

The manuscript is clear and very innovative, however the authors should better discuss the need for further research to elucidate specific miRNA-mRNA interactions and their functional consequences in the context of FASD.

It would also be interesting to add more information on potential therapeutic targets or strategies to mitigate the effects of prenatal alcohol exposure on the hippocampus and improve outcomes for people with FASD.

Author Response

We thank this review for excellent suggestions.

We have added a paragraph to the discussion outlining some of the attempts to improve the lives of people living with FASDs.  Of high relevance both to the topic of this paper and to global health, recent studies have show miRNA profiles to be consistently altered in individuals with FASD and with individuals likely to give birth to offspring with FASD.  As these profiles can be captured relatively non-invasively (in blood, for example) they have high relevance as possible biomarkers that could be used both predictively and therapeutically.  

Reviewer 2 Report

The manuscript is nicely done and written. The study design is appropriate and apparently, the analyses were carefully performed.  I believe that the results are valuable for the scientific community and has significant scientific merit, as it will probably ignite many further studies in the near future.

However, some points need to be clarified before the publication.

General remark – manuscript style must follow journal guidelines. Line numbers, headlines etc. are missing.

Page 2 - From histological point of view there are only four kinds of tissues: epithelial, connective, muscular and nervous. Therefore, hippocampus is not a tissue, it is nervous structure. The term “hippocampal tissue” (Page 3) is not justified.

Page 2 – it is not clear how many animals (n=?) were subjected to the study.

Page 3 – please describe more carefully a method of euthanasia (does etc.)

Page 4 - what post hoc-test was used for ANOVA?

Page 14 – please add any conclusions

Page 15 – reference 1 is missing

Author Response

We thank this reviewer for attention to detail and for very helpful comments

With respect to the remark about journal guidelines, it seems there was a minor problem in uploading the correct version:  this has now been corrected.  

Page 2:  you are of course correct.  We have revised the sentence to refer to the hippocampus as a structure and to remove all mentions of "hippocampal tissue."

Page 2:  We have added two sentences to the Statistical Analysis specifying the total number of animals used, and the number of animals in each group.

Page 3:  We have provided details of the method of euthanasia.  

Page 4:  In this study, we used a 2x2 ANOVA which involved two independent variables (PAE, Age) each with two levels (case/control and 5 months/2 years) and the possibility of measuring the interaction effect. For this main effects, post hoc analysis was not used given that there are only two levels to the data.

Regarding the interaction effect, there was a post hoc analysis in the sense that we generated interaction plots (supplemental information) in order to determine if there was a consistency amongst these interaction plots that might underpin a biological cause for the interaction. This avenue did not produce a consistency that could be tied to a biological effect and was not explored further.

Page 14:  We have added a "Conclusions" secction

Page 15:  I have verified that reference 1 is visible.